# Evolution and activation mechanism of the flavivirus class II membrane-fusion machinery

Marie-Christine Vaney [1,8], Mariano Dellarole [1,5,8], Stéphane Duquerroy[1,2,8], Iris Medits[3], Georgios Tsouchnikas[3,6], Alexander Rouvinski[1,7], Patrick England [4], Karin Stiasny [3✉], Franz X. Heinz [3✉] & Félix A. Rey [1✉]

The flavivirus envelope glycoproteins prM and E drive the assembly of icosahedral, spiky immature particles that bud across the membrane of the endoplasmic reticulum. Maturation into infectious virions in the trans-Golgi network involves an acid-pH-driven rearrangement into smooth particles made of $(prM/E)_2$ dimers exposing a furin site for prM cleavage into "pr" and "M". Here we show that the prM "pr" moiety derives from an HSP40 cellular chaperonin. Furthermore, the X-ray structure of the tick-borne encephalitis virus $(pr/E)_2$ dimer at acidic pH reveals the E 150-loop as a hinged-lid that opens at low pH to expose a positively-charged pr-binding pocket at the E dimer interface, inducing $(prM/E)_2$ dimer formation to generate smooth particles in the Golgi. Furin cleavage is followed by lid-closure upon deprotonation in the neutral-pH extracellular environment, expelling pr while the 150-loop takes the relay in fusion loop protection, thus revealing the elusive flavivirus mechanism of fusion activation.

[1] Institut Pasteur, Université Paris Cité, CNRS UMR 3569, Unité de Virologie Structurale, Paris, France. [2] Université Paris Saclay, Faculté des Sciences, Orsay, France. [3] Center for Virology, Medical University of Vienna, Vienna, Austria. [4] Institut Pasteur, Université Paris Cité, CNRS UMR 3528, Plateforme de Biophysique Moléculaire, Paris, France. [5] Present address: CIBION, CONICET, Buenos Aires, Argentina. [6] Present address: HOOKIPA Pharma 19 Inc, Vienna, Austria. [7] Present address: Department of Microbiology and Molecular Genetics, Institute for Medical Research Israel-Canada, The Kuvin Center for the Study of Infectious and Tropical Diseases, The Hebrew University of Jerusalem, Jerusalem, Israel. [8] These authors contributed equally: Marie-Christine Vaney, Mariano Dellarole, Stephane Duquerroy. ✉email: karin.stiasny@meduniwien.ac.at; Franz.X.Heinz@meduniwien.ac.at; felix.rey@pasteur.fr

Flaviviruses include pathogens that cause a high public health burden worldwide, such as dengue viruses (DENV1 to DENV4)[1], Zika virus (ZIKV)[2], Japanese encephalitis virus (JEV)[3], West Nile virus (WNV)[4] or yellow fever virus (YFV)[5], which are transmitted by mosquitoes, and tick-borne encephalitis virus (TBEV)[6], Kyasanur Forest Disease virus (KFDV)[7] or Powassan virus (POWV)[8] transmitted by ticks. Structural studies have shown that the flavivirus envelope protein E, responsible for driving membrane fusion, has homologs in many different enveloped viruses - termed class II enveloped viruses - belonging to otherwise unrelated families, such as the alphaviruses[9], rubella virus[10] and the members of several families of bunyaviruses[11–15]. We note, however, that in the *Flaviviridae* family, only the viruses in the *Flavivirus* genus have a class II fusion protein; those in the other three genera (*Hepacivirus*, *Pegivirus* and *Pestivirus)* display unrelated envelope proteins[16]. The class II fusion protein fold is rich in β-sheets, with three characteristic domains, termed I, II and III exposing an internal "fusion loop" at the distal tip of domain II[17]. The fusion loop must insert into the target membrane upon low pH triggering to initiate membrane fusion[18].

The flavivirus particle's life cycle includes three key steps: 1- budding into the neutral pH environment of the ER lumen; 2- transport across the trans-Golgi network (TGN), where the pH is mildly acidic and where the particles undergo a maturation step before exiting to the neutral pH extracellular environment;[19,20] and 3- entry via receptor-mediated endocytosis into a target cell, where the mildly acidic endosomal pH triggers E-driven fusion of the viral envelope with the endosomal membrane. The fusogenic conformational change of E is irreversible, and the consecutive steps outlined above require a special particle maturation mechanism in order to avoid premature fusion-triggering in the acidic TGN during exocytosis. Although other class II enveloped viruses also bud in the ER, such as members of the *Phenuivirus* family in the *Bunyavirales* order[21], their life cycle involves fusion in late endosomes of the target cell, which have a more acidic pH than the TGN. They therefore are not sensitive to the mildly acidic pH of the TGN and do not require a maturation process.

The immature flavivirus particles that bud into the ER lumen display prM/E protomers in which the pr moiety binds at the domain II tip to cap the fusion loop, avoiding its insertion into the ER membrane[22]. These protomers form 60 (prM/E)₃ trimeric spikes in a head-to-head orientation and interconnected within an icosahedral surface lattice[23,24]. Subsequent maturation into infectious particles during transport to the cell surface involves an acid-pH-induced reorganization of the 180 prM/E protomers at the particle surface. From forming 60 trimeric spikes, the protomers rearrange to form 90 head-to-tail dimers interacting laterally to make a smooth particle with an icosahedral herringbone-like arrangement[25,26]. Cleavage by the TGN-resident furin protease then takes place at a prM site that becomes exposed in the (prM/E)₂ dimers[27,28] to yield proteins pr (N-terminal half) and M, the membrane-anchored C-terminal half of prM. pr is shed from the particle upon subsequent secretion into the extracellular environment, leaving an activated particle, prone to mediate acidic pH-triggered membrane fusion upon entry into a target cell.

A mechanistic molecular understanding of the pH-driven particle transitions taking place during maturation and secretion of flavivirus particles is lacking. Here we describe the X-ray structure of the soluble E (sE) dimer of TBEV in complex with pr at acidic pH, revealing a crucial role for the 150-loop of E domain I in relaying the fusion-loop capping role of pr upon secretion. Our analyses further show that pr - unlike the accompanying protein observed in the ancestral membrane fusion machinery seen in other class II viruses - is related to the family of DnaJ/HSP40 cellular co-chaperons (or chaperonins) present in the ER,

reflecting a specific adaptation to the unique features of flavivirus morphogenesis and entry into cells.

## Results

**pr stabilizes the sE dimer at acidic pH**. Size-exclusion chromatography (SEC) combined with multi-angle light scattering (MALS) revealed a pH dependent interaction between purified sE and pr. At pH 8, sE eluted as dimer and pr as monomer (Fig. 1a, top panel). At pH 5.5, the sE dimer dissociated into monomers while pr remained unaltered (Fig. 1a, middle panel). The sE monomer elutes late from the SEC column, in fractions normally corresponding to the elution of small molecules and not of proteins of its molecular mass (∼50 kDa). The reason is most likely retention in the column by interactions of the fusion loop with the resin, as observed previously with other class II fusion proteins[29]. The mixture of sE with a pr excess at pH 5.5 had an elution pattern (Fig. 1a, bottom panel) similar to that observed with the two proteins assayed separately at pH 8 (Fig. 1a, top panel). For clarity, the chromatograms obtained by running sE incubated with a pr excess at pH 8 and 5.5 are shown superimposed in Fig. 1b, with the indicated fractions analyzed by SDS-PAGE in Fig. 1c. Although the mass estimation for the sE fraction by MALS did not show an increase indicating that pr was bound to the sE dimer at acidic pH (Fig. 1a, bottom panel), SDS-PAGE clearly showed that pr binds to the sE dimer at pH 5.5 but not at pH 8 (Fig. 1b, c). Most importantly, these experiments demonstrated that pr restores sE dimerization at pH 5.5, since no sE monomer was detected when pr is in excess at this pH.

To measure the pr affinity for sE as a function of pH, we used surface plasmon resonance (SPR) with pr immobilized in the SPR chip and flowing sE at different concentrations. We scanned from pH 5 to 8 by 0.5 pH units, and observed a sharp increase with pH of the estimated $K_D$ values, which ranged from 66 nM at pH 5.5 to 10 μM at pH 7.5 (Fig. 1d; Supplementary Fig. 1; Supplementary Table 1), with the highest increase observed between pH 6.5 and 7 (Fig. 1e). Figure 1e shows a right-ward shift of the curves from pH 5.0 to 6.5, suggesting modest conformational or charge changes to the affinity, yet they reach a common saturation level. Above 6.5 (the protonation pH of histidine), the association rate is drastically reduced and therefore there is saturation at a much lower level, indicating that the binding surface might not exist under those conditions despite the presence of an sE dimer at neutral pH. In conclusion, the drastic drop in binding affinity upon raising the pH in the explored range suggests an electrostatic and/or a conformational change at the sE-pr interface upon deprotonation.

**X-ray structure of the (pr/sE)₂ dimer**. To further confirm the presence of pr bound to sE as detected by SDS-PAGE at pH 5.5 (Fig. 1b, c), we submitted the corresponding SEC fraction to crystallization trials. Crystals grew at pH 4.6, which diffracted to 2.6 Å resolution (Supplementary Table 2). The X-ray structure, determined by molecular replacement (see Methods), showed two pr molecules bound symmetrically at the sE dimer interface (Fig. 2), each contacting the tip of domain II of one sE subunit and segments of domains I, II and III of the opposite sE subunit in the dimer (Fig. 2a, b). The pr contacts with the tip of domain II are similar to those described previously for the monomeric DENV2 pr/sE complex[30] and to those observed in the spiky immature flavivirus particles at neutral pH[23,24], thereby identifying the pr/sE protomer in the dimer. The contacts of pr with the domain II tip are therefore intra-protomer, while those with domains I, II and III of the other sE subunit, which had not been observed previously, are inter-protomer. The buried surface area (BSA) per pr binding site on the sE dimer is 1265 Å², 62% of

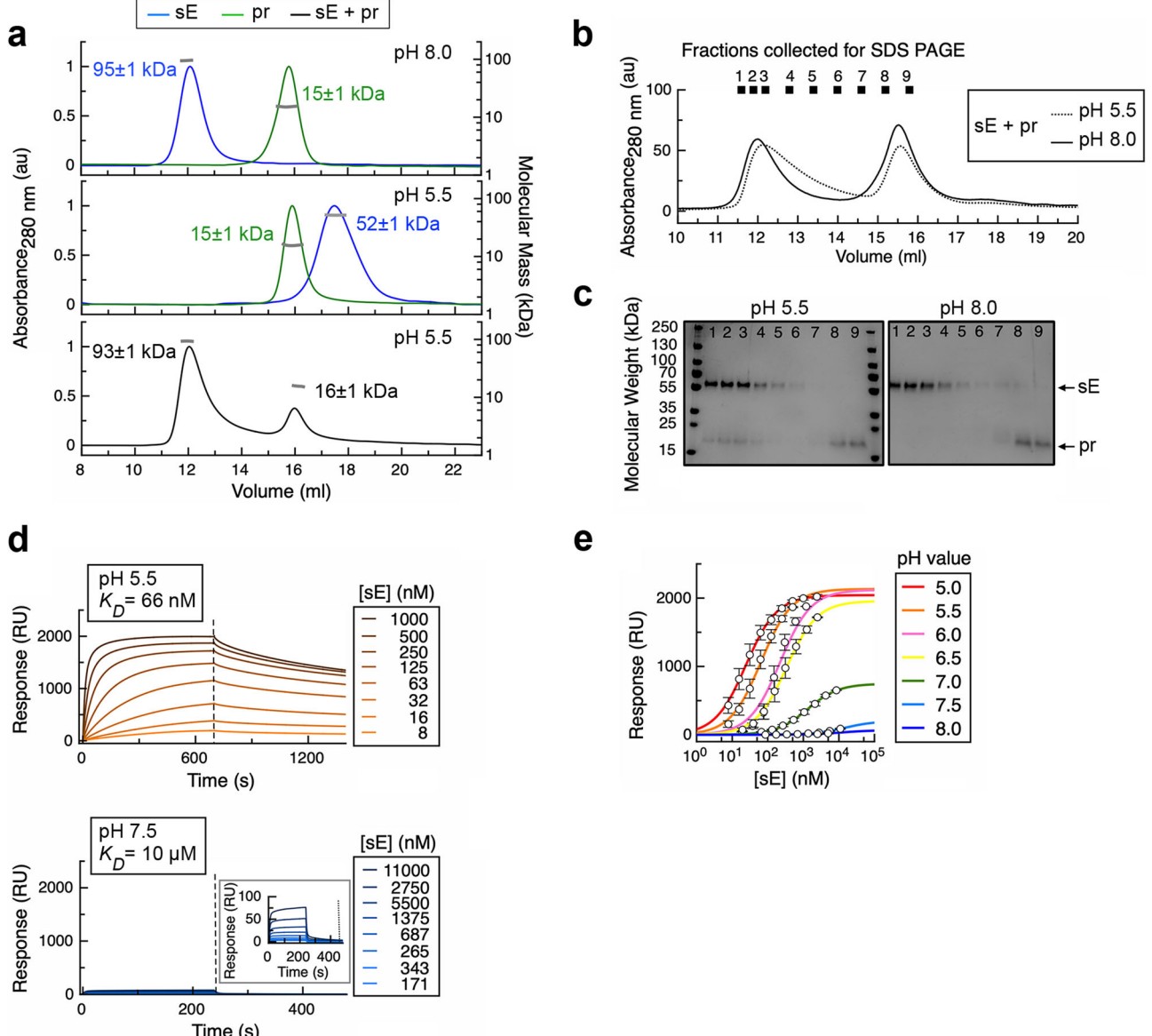

**Fig. 1 pr stabilizes the sE dimer at acidic pH. a** SEC elution volume profiles of isolated sE and isolated pr equilibrated at pH 8.0 (top panel) and at pH 5.5 (middle panel). The bottom panel shows the elution of a mixture of sE with pr in excess (1:2 sE:pr monomer:monomer molar ratio) at pH 5.5. Left y axis: the ultraviolet absorbance normalized by setting the highest peak to 1. Right y axis: molecular mass (kDa) determined by MALS, with the values for each species indicated on the corresponding peak. **b** Superposition of the SEC elution profiles of the mixture of sE with an excess pr equilibrated at pH 8.0 (smooth curve) and at pH 5.5 (dotted curve) as described in Methods. The fractions analyzed by SDS-PAGE in (**c**) are indicated (1–9). **c** SDS-PAGE of the SEC fractions indicated in (**b**) at pH 5.5 (left) and pH 8 (right); Coomassie blue staining. **d** Surface plasmon resonance (SPR) sensorgrams with pr immobilized on the SPR chip and sE flowed at different concentrations (color-coded as indicated) at pH 5.5 (top panel) and pH 7.5 (bottom panel). Both panels are shown at the same vertical scale (0–2000 RU), with the inset in the bottom panel amplifying the range between 0 and 100 RU to show the actual curves, as the interaction at pH 7.5 is much weaker. A dashed vertical line indicates the transition from association to dissociation kinetics. **e** SPR association equilibrium values as a function of sE protein concentration for the titrations measured at different pH values, ranging from 5 to 8. The measurements were reproduced twice. Error bars display the mean ± SD for each individual data point. Color lines fits a single-site interaction model. Source data are provided as a Source Data file.

which corresponding to intra-protomer and the remainder to inter-protomer contacts (Fig. 2b). On pr, the surface buried by the sE dimer is 1328 Å$^2$, of which about 80% correspond to the "capping loop" (CL) (labeled in Figs. 2a, 3a), so-named because it caps the E fusion loop, as described in more detail below. Mapping the amino-acid conservation to the pr and sE surfaces showed that both interacting regions are highly conserved across tick-borne flaviviruses (Fig. 2c; Supplementary Fig. 2), although there is variability when comparing to mosquito-borne viruses (Supplementary Fig. 3). Moreover, the surface electrostatic

potential of sE and pr are highly complementary at pH 5.5 but not at 7.5, because of a specific deprotonation of the E protein (Fig. 2d).

## pr is derived from HSP40/DnaJ chaperonins of the host. The first hits of a DALI[31] search with the TBEV pr atomic model corresponded, as expected, to pr of other flaviviruses, with the highest Z score (15.5) for pr from yellow fever virus (Table 1). Lower but significant Z scores were also obtained with the

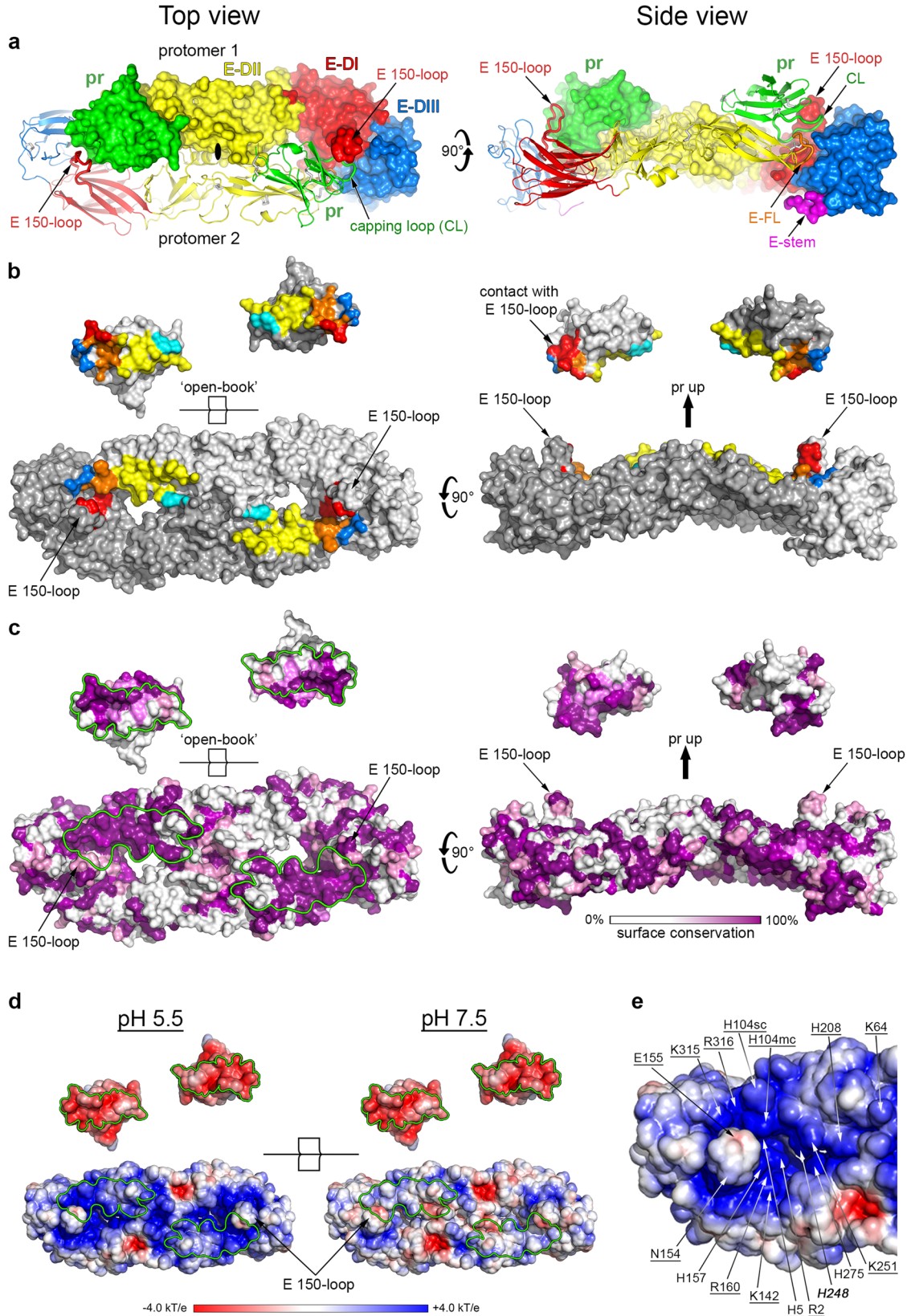

substrate binding domain (SBD) of type I and II HSP40/DnaJ protein cofactors (or chaperonins) of the heat shock protein 70 (HSP70) cellular chaperone system[32]. In the HSP40 molecules, the SBD is composed of two related consecutive β-sandwich sub-domains termed "substrate binding motifs 1 and 2" (SBM-1 and

SBM-2, Fig. 3b, right panels) located between an N-terminal J domain and a small C-terminal α-helical dimerization domain[32]. SBM-1 and 2 appear to have arisen by gene duplication and share a unique folding topology in the CATH fold database (CATH Topology level 2.60.260: "HSP40/DnaJ peptide-binding domain").

**Fig. 2 X-ray structure of the (pr/sE)₂ dimer. a** The (pr/sE)₂ dimer shown in two orthogonal views, with E color-coded by domains as indicated (E domain I red, II yellow, III blue, stem magenta and fusion loop (E-FL) in orange). pr is colored green. One pr/sE protomer is shown in ribbons and the other in surface representation, with a central solid black oval in the top view marking the crystallographic 2-fold symmetry axis. Elements such as the E fusion loop (FL), E 150-loop, E stem (residues 396–400 downstream domain III) and pr capping loop (CL) discussed in the text are labeled. **b** pr footprint on the sE dimer surface. The two protomers are shown in two shades of grey in an open-book representation (left panel), and with the pr subunits shifted up in side view (right panel). The buried surfaces in the complex are colored according to the E domains involved, as in (**a**), except that the inter-protomer contacts with domain II are colored cyan instead of yellow. **c** The same surface as in (**b**) heat colored as indicated in the bar underneath to highlight amino acid conservation at the pr and sE dimer surface across tick-borne flaviviruses infecting vertebrates (from the alignment shown in Supplementary Fig. 2). The buried surfaces of E and pr in the complex are outlined in green in the left panel. Note that the interaction surface is the most conserved region of both, pr and E. **d** Same view as in the left panels in (**b**) and (**c**), colored according to the electrostatic surface potential (as indicated in the color-code bar) computed at pH 5.5 (left panel) and at pH 7.5 (right panel). The interaction surfaces are outlined in green. **e** Closeup of the electrostatic potential of the sE dimer surface of interaction with pr at pH 5.5, showing numerous charged residues (labeled). Residues conserved in tick-borne flaviviruses are underlined, and those conserved in all flavivirus are in bold and italic. The glycosylated Asn154 is also labeled.

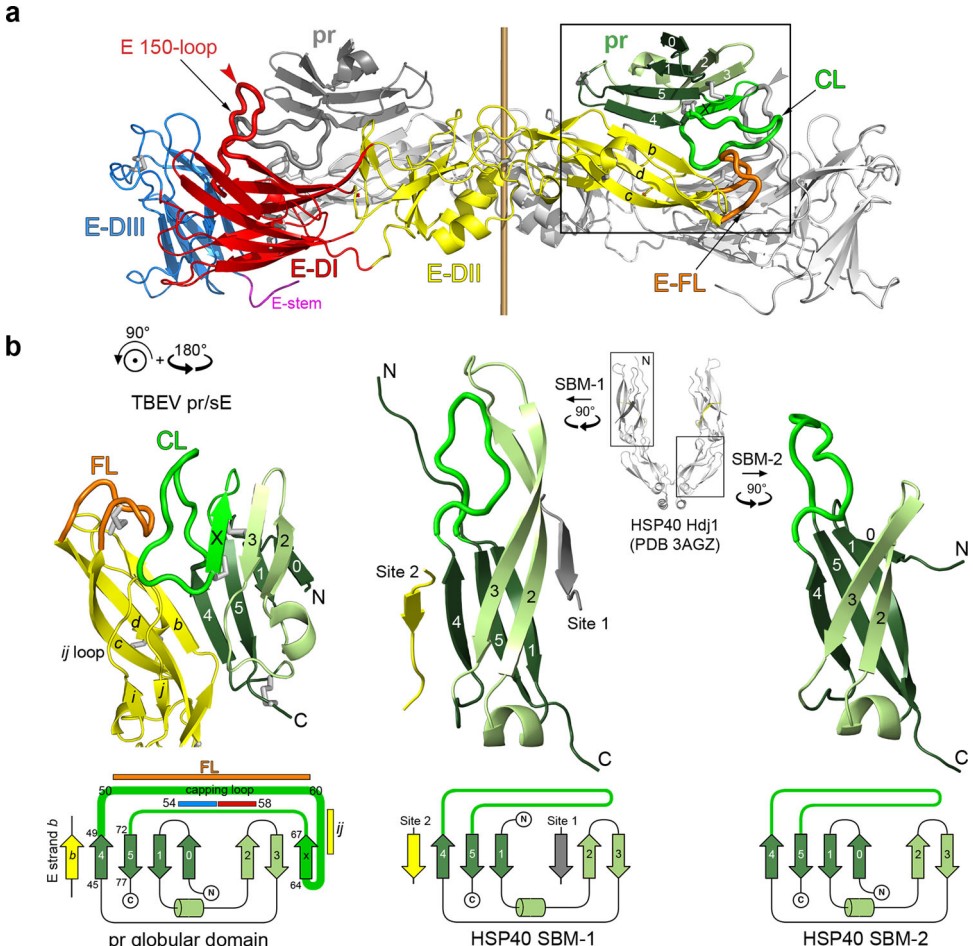

**Fig. 3 pr is a homolog of the substrate binding domains of HSP40/DnaJ chaperonins. a** The (pr/sE)₂ dimer shown in side view as ribbons, with one protomer in colors as in Fig. 2a and the other in grey. The various domains are labeled, and a central bar indicates the crystallographic 2-fold axis of the complex. A red arrowhead marks the location of glycosylated Asn154 in the 150-loop. **b** The framed region in (**a**) is shown in the left panel, rotated as indicated, to match the orientations of the SBM-1 and −2 domains of the human HSP40 Hdj1 chaperonin shown in the middle and right panels, respectively. An inset in between the middle and right panels shows the X-ray structure from which the SBMs were extracted, and corresponds to the Hdj1 (PDB 3AGZ)[66] HSP40 chaperonin dimer with two HSP70-bound peptides. The rotations applied to the SBM-1 (middle panel) and SBM-2 (right panel) are indicated. The pr capping loop (CL; thicker line, bright green) corresponds to an extension of the β4β5 loop in the SBMs, which is postulated to be involved in interactions with substrate. In pr, the extension leads to an additional strand, labeled βₓ. The two peptides bound to SBM-1 are indicated in grey (site 1) and yellow (site 2), with the color emphasizing the correspondence between the pr/sE association and site 2 binding through main-chain β interactions. Different shades of green are also used to highlight equivalent regions of pr and the SBMs, displayed in the same shades. The topological organization of the displayed beta sandwiches is shown at the bottom of each panel. Colored bars on the topology diagram of pr indicate the regions of interaction with E, color-coded as defined in the legend to Fig. 2a. Residue numbers along the pr topology diagram are a guide to indicate residues forming β-strands and segments of the capping loop interacting with the indicated regions of E.

**Table 1 List of structural neighbors of TBEV pr found by the Dali server[31, 65] sorted by Z-score. Only matches with Z-score > 2.8 are listed.**

| PDB-chain | Z | rmsd | lali | nres | %id | Description |
|---|---|---|---|---|---|---|
| 6epk-B | 15.5 | 0.9 | 79 | 80 | 29 | YFV pr – X-ray structure |
| 3c5x-C | 12.4 | 1.6 | 77 | 81 | 25 | DENV2 pr X-ray structure |
| 5u4w-F | 12.5 | 1.6 | 77 | 81 | 23 | ZIKV pr from the cryo-EM structure of the immature virion |
| 6zqi-B | 11.7 | 1.3 | 78 | 101 | 36 | SPOV pr from the cryo-EM structure of the immature virion |
| 7l30-b | 11.5 | 1.5 | 74 | 150 | 42 | BINJV pr from the cryo-EM structure of the immature virion |
| 2b26-B | 3.5 | 2.7 | 53 | 147 | 6 | Yeast HSP40-SIS1- fragment C-terminal domain |
| 3agx-B | 3.5 | 3.1 | 47 | 148 | 6 | Human HSP40-Hdj1 peptide binding domain |
| 1xao-A | 3.4 | 2.7 | 50 | 115 | 6 | Yeast HSP40-Ydj1 dimerization domain |
| 7jtk-Y | 3.2 | 2.6 | 52 | 213 | 10 | *Chlamydomonas ReinhardtII* HSP40 - flagellar radial spoke protein |
| 3agz-A | 3.2 | 2.7 | 53 | 185 | 9 | Human HSP40-Hdj1 peptide binding domain |
| 2q2g-A | 3.2 | 2.8 | 54 | 176 | 11 | Cryptosporidium parvum HSP40 dimerization domain |
| 6jzb-A | 3.2 | 2.8 | 51 | 251 | 6 | *Streptococcus Pneumoniae* Type I HSP40 - DnaJ |
| 1nlt-A | 3.1 | 2.8 | 52 | 228 | 6 | Yeast HSP40-Ydj1 |
| 1c3g-A | 3.0 | 2.5 | 51 | 170 | 6 | Yeast HSP40-Sis1 - C-terminal peptide binding domain |
| 4j80-A | 2.9 | 2.7 | 50 | 271 | 12 | *Thermus Thermophilus* Chaperone protein DnaJ-2 |

Although the similarity had not been noticed when the first X-ray structure of DENV2 pr was reported[30], the CATH server has automatically assigned flavivirus pr as one of three superfamilies sharing this particular topology (http://www.cathdb.info/browse/sunburst?from_cath_id=2.60.260). The SBMs feature a main $\beta$-sheet ($\beta_4\beta_5\beta_1$, Fig. 3b, middle and right panels) packing against a $\beta$-hairpin ($\beta_2\beta_3$). The $\beta$-strand topology of pr (Fig. 3b, left panel) is closest to that of the SBM-2 module, which has an additional strand in the main $\beta$-sheet with respect to SBM-1 (labeled $\beta_0$, Fig. 3b, right panel). The SBMs display a surface hydrophobic patch and have been crystallized in complex with HSP70-derived peptides bound to SBM-1. The visible portion of the bound peptides make additional side $\beta$-strands, one interacting anti-parallel to strand $\beta_2$ (site 1) and the other running antiparallel to $\beta_4$ (site 2). Both SBMs feature a prominent loop connecting $\beta_4$ and $\beta_5$, which contributes to the hydrophobic surface of the SBMs used to accommodate partially folded protein substrates. In pr, this loop corresponds to the "capping loop" introduced above (Figs. 2a, 3). pr features an additional insertion with respect to the SBMs at the end of the capping loop, which makes the $\beta_X$ strand, thereby turning the hairpin into a three-stranded $\beta$-sheet ($\beta_X\beta_3\beta_2$; Fig. 3b, left panel). In the complex with E, $\beta_4$ makes a parallel interaction with $\beta$-strand $b$ of E domain II (Fig. 3b, left panel), creating a single 7-stranded $\beta$-sheet across both subunits of the pr/sE protomer.

**Interactions between pr and the sE dimer**. In the available structure of the TBEV sE dimer at neutral pH[17] (PDB 1SVB), the 150-loop (residues 146–160, connecting the adjacent $E_0$ and $F_0$ $\beta$-strands of the domain I $\beta$-sandwich) buries a 1-turn helix at the E N-terminus (termed N-ter helix) within domain I, in an intra-chain interaction. At the same time, it makes an inter-chain interaction with the fusion loop of the other subunit of the sE dimer (Fig. 4a, right panel). Comparison with the structure of the (pr/sE)$_2$ complex shows that the 150-loop acts as a hinged lid, which opens at acidic pH (Fig. 4a, b, left panel) and closes when the pH is neutral (Fig. 4a, b, right panels; curved arrows) owing to electrostatic repulsion with the buried N-ter helix upon proto-nation (Fig. 2e), concomitantly expelling pr from the complex. At acidic pH, the pr capping loop fits snugly into the positively charged cavity formed upon lid opening (Fig. 2d, e), with intra-protomer contacts with the E $ij$ hairpin and fusion loop on one side (Fig. 3b, left panel; Supplementary Fig. 4) and inter-protomer interactions with E domains I and III on the other (Fig. 4a, b, left panels). pr thus wedges in at the sE dimer interface, in line with

its role in stabilization of the sE dimer at acidic pH (Fig. 1). In addition, the $fg$ loop (domain II) is displaced to interact with the $ij$ hairpin (also domain II) across the sE dimer interface, and with pr segments in strands $\beta_4$ and $\beta_5$ (Fig. 4c).

The list of all the interactions observed between pr and sE is provided in Supplementary Table 3. The polar and electrostatic bonds include 20 intra-protomer bonds with the tip of domain II and 13 inter-protomer bonds with domains I, II and III on the adjacent sE subunit (Supplementary Table 3). The interactions of the capping loop make a highly interconnected network of polar bonds at the E dimer interface (Fig. 4; Supplementary Table 3). Although the crystal had interpretable electron density for the Asn154 side chain in the 150-loop, the attached glycan was disordered (Supplementary Fig. 5), contrary to the sE dimer at pH 8, in which the glycan interacts across the sE dimer interface[17] (Fig. 4a, right panel).

**Conformational rearrangement of the 150-loop upon pr dissociation at neutral pH**. There are several specific structural changes in sE in the complex with pr at low pH with respect to the previous structure of sE at pH 8 (compare left and right panels in Fig. 4; see also Supplementary Table 4). The main one is the conformation of the 150-loop (Fig. 4a, b), which is accompanied by small shifts in the N-ter helix (His5; Fig. 4a), the $fg$-loop (His208; Fig. 4c), and the fusion loop across the dimer interface (His104; Fig. 4a, b). The 150-loop, which adopts an "open lid" vertical conformation in the presence of pr (Fig. 4a, b, left panels), switches to a "closed lid" horizontal orientation upon pr release at neutral pH (Fig. 4a, b, curved arrows), contacting residues Gly102 and His104 of the fusion loop across the E dimer interface (Fig. 4a, b, right panels), helping keep it in place in the absence of pr at neutral pH. It thus acts as a snap-lock to maintain the dimer in a conformation primed to react and drive fusion upon subsequent exposure to low pH.

**The 150-loop and the E protein N-terminus in different flaviviruses**. The 150-loop of the flavivirus E protein, displayed in Supplementary Fig. 6, varies in length between 13 and 21 residues, as shown in the alignment shown in Supplementary Fig. 3. All the available structures show the positively charged N-terminus of E buried beneath the 150-loop (Supplementary Fig. 6), as well as a cluster of histidine residues in the region (Supplementary Fig. 7). The open-book representation of Fig. 5 shows that in all cases, the buried face of the 150-loop becomes protonated and positively charged at acid pH but not at neutral

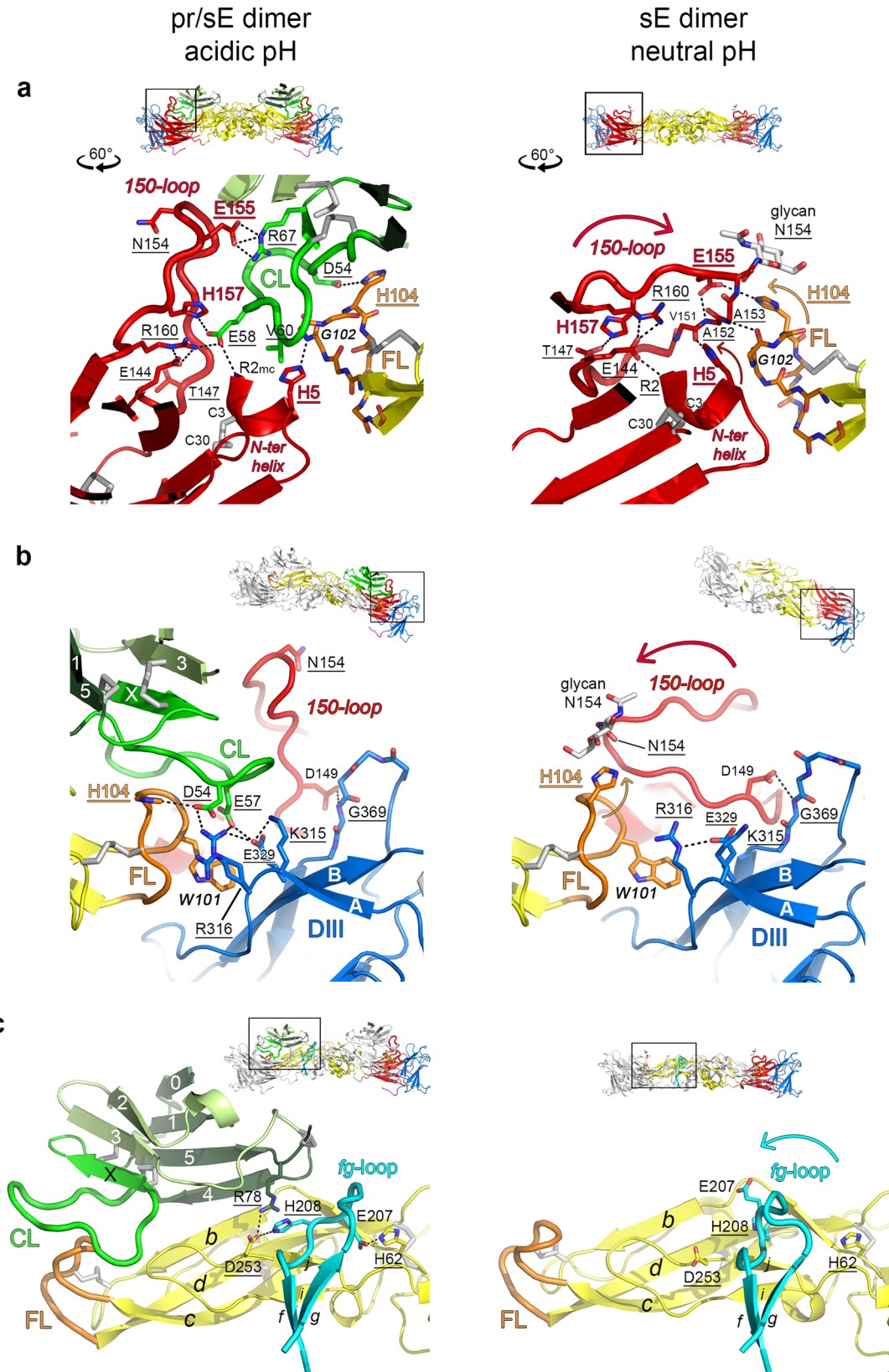

pH. Therefore, despite an important sequence variability (Supplementary Fig. 3) compared to the rest of the E protein, the functional role of the 150-loop as a hinged-lid that opens and closes depending on the environmental pH is not specific to tick-borne flaviviruses but also applies to the medically highly relevant mosquito-borne flaviviruses.

Inspection of the available DENV2 pr/E structure (PDB 3C5X) revealed a disordered 150-loop. This structure was obtained using a single-chain construct connecting the C-terminus of the prM ectodomain (prMe) to the N-terminus of sE via a flexible linker[30], thereby eliminating the constitutive positive charge buried at the site corresponding to the E N-terminus. Furthermore, the crystals

**Fig. 4 Structural changes in the E dimer upon pr ejection at neutral pH.** A box on the ribbon diagrams at the top of each panel show the closeup region, with black symbols denoting the rotation applied. **a** Lid-closing. At acidic pH (left panel), pr wedges in at the E dimer interface, inserting the capping loop (CL) in between the fusion loop (orange, labeled) of one E subunit and the 150-loop (red) of the other, making multiple electrostatic bonds as labeled. At neutral pH (right panel), deprotonation eliminates the intra-subunit electrostatic repulsion between the N-ter helix and the 150-loop. The overall effect is pr ejection and 150 loop lid-closing (large red arrow), which now interacts directly with the fusion loop across the E dimer interface. The Asn154-linked glycan then stacks against the His104 side chain, which switches rotamer to accommodate this interaction (orange arrow). The N-ter helix also rotates slightly to allow new interactions between His5 and the main chain of Arg2 with the 150-loop lid (small red arrow). Glu155 and His157 in the 150-loop undergo a large change in location to make alternative polar interactions. **b** Involvement of domain III. Several charged side chains of domain III (Arg316, Lys315, Glu329) participate in the cluster of inter-protomer electrostatic interactions with pr, mainly involving Asp54 and Glu57 of the capping loop, which also make intra-protomer interactions with protonated His104 of the fusion loop (left panel). Asp149 in the 150-loop, located near the base of the hinge-lid, undergoes an important change in its main chain, yet maintains its side chain hydrogen-bonded to the domain III main chain at Gly369. **c** Shift of the *fg*-loop in domain II. The left panel shows that at acidic pH, the protonated side chain of His208 in the *fg*-loop (cyan) makes an inter-protomer salt bridge with Asp253 in the *ij* hairpin, which makes an intra-protomer salt bridge with Arg78 of pr. In addition, Glu208 of the *fg*-loop makes an inter-protomer salt bridge with the protonated His62 in domain II. These interactions are lost at neutral pH and in the absence of pr, leading to a different conformation of the *fg*-loop (right panel), which moves as shown by the cyan arrow.

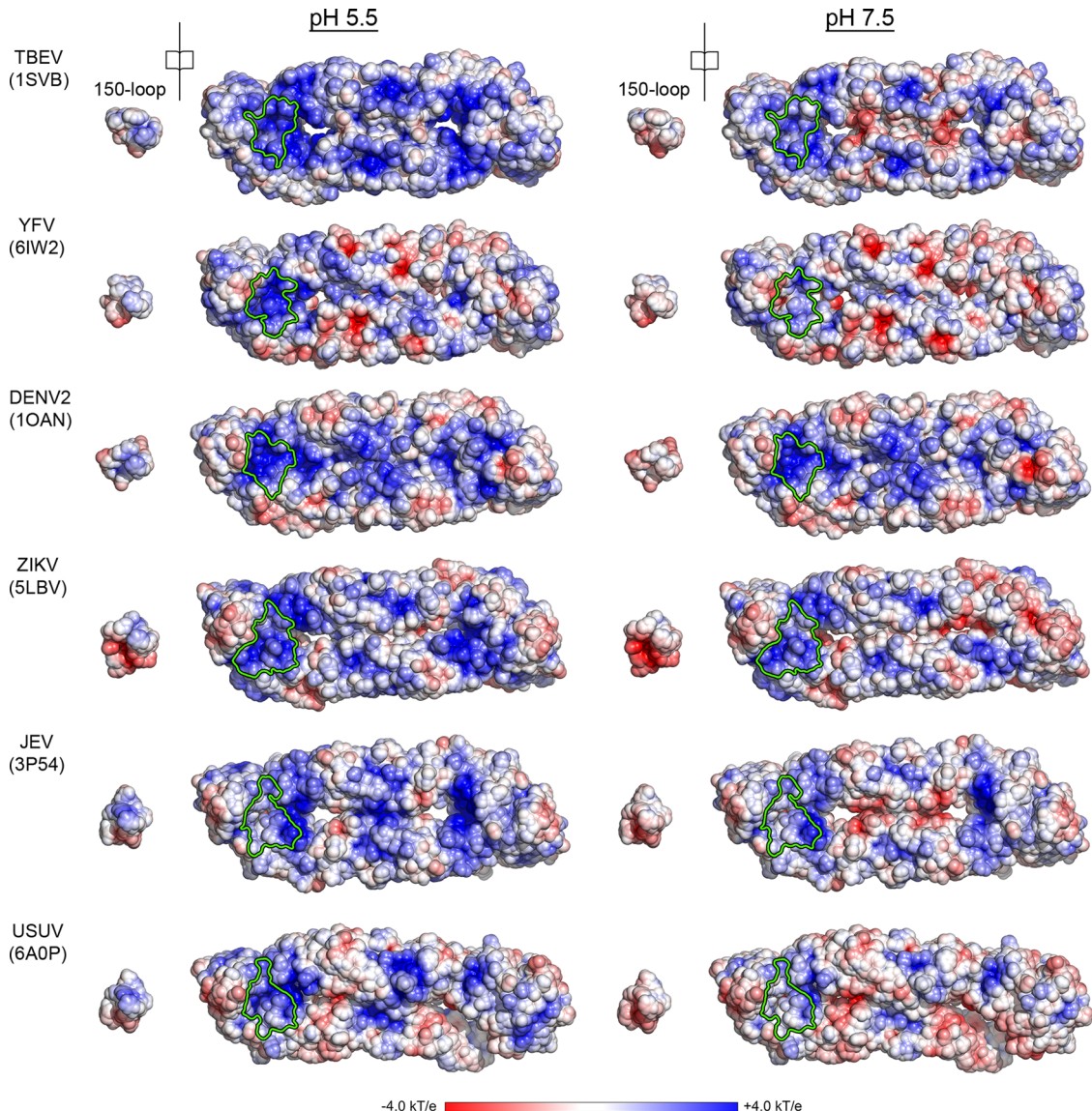

**Fig. 5 Specific protonation of the 150-loop at acid pH across flaviviruses.** The various X-ray structures of the flavivirus sE dimer shown were determined from crystals grown at pH above 7.5. They are labeled with the corresponding virus and PDB code, and shown in surface representation colored according to their surface electrostatic potential (as indicated in the color code bar) calculated at pH 5.5 (left panels) and pH 7.5 (right panels). The 150-loop was extracted from the left half of the sE dimer and is shown in an open book representation to show the protonation state of its buried surface (i.e., the lid's inner surface) at pH 5.5 and 7.5. The matching surface buried by the lid in the sE dimer is outlined in green. This surface includes the authentic N-terminus of E (see also Supplementary Fig. 6).

showed clear density for the artificial linker, allowing the authors to partially build it extending from the E N-terminus, as illustrated in Supplementary Fig. 8. We initially used a similar construct for TBEV prM/E, albeit with a different linker connecting the ectodomains (Supplementary Fig. 8a; see Methods). The single-chain TBEV prMe-linker-sE protein behaved as monomer in solution (Supplementary Fig. 8b) but crystallized as a dimer (Supplementary Table 2). Its X-ray structure (Supplementary Fig. 8c) also showed a disordered 150-loop and a partially ordered linker extending from the E N-terminus (Supplementary Fig. 8d, left panel), but with pr still bound at the sE dimer interface. In short, in both the DENV2 and the TBEV structures, the presence of the artificial linker prevented ordering of the 150-loop (Supplementary Fig. 8d). The difference was that the DENV2 prMe-linker-sE structure showed a monomer, while the TBEV counterpart is a dimer. We therefore tested TBEV sE dimer destabilization by mutating Trp101 of the fusion loop, which makes important interactions at the sE dimer interface (see Supplementary Fig. 7), into aspartic acid (Supplementary Fig. 8a, right panel). The prMe-linker-sE$^{W101D}$ construct crystallized at low pH with sE in the post-fusion trimer-of-hairpins conformation instead of a dimer (see Supplementary Table 2 for the crystallographic statistics), with pr bound to the sides of the trimer (Supplementary Fig. 8e) via the same intra-protomer contacts with domain II observed in the $(pr/sE)_2$ complex. These results indicate that not only lid-opening, but also the correct E-E interactions within the dimer, as provided by Trp101 in the fusion loop (Fig. 4b; Supplementary Fig. 7), are required for the inter-protomer pr/E interactions observed in the $(pr/sE)_2$ structure. They further show that the mechanism of fusion protection by pr in the TGN does not involve interfering with post-fusion E trimer formation, since the trimer can form with pr bound, but rather by E dimer stabilization at low pH to bury the fusion loop and drive the formation of smooth immature particles with a herringbone pattern of dimers in the TGN.

## Discussion

Our data provide mechanistic insight into two essential steps of flavivirus maturation and reveal a key role of the E 150-loop as a protonation-gated hinged lid. Although our studies used TBEV, the analyses presented in Fig. 5 and in Supplementary Figs 6 and 7 indicate that this mechanism is universal across flaviviruses. We show that the final stage of flavivirus maturation is not just a passive, pH-dependent "pr shedding" at neutral pH, but that it is induced by a switch of the hinged lid from an open to a closed conformation that expels pr from the particle. Indeed, the intra-protomeric pr/E interactions are stable at neutral pH, at least stable enough in the context of the prM/E protomer to make the spiky immature particles in the ER. In the context of the E dimer, lid closure is incompatible with the presence of pr bound, with the interactions of the 150-loop in the closed-lid conformation at neutral pH involving stronger bonds than those maintaining the pr/E protomer. Furthermore, the 150-loop in the closed-lid conformation takes the relay from pr by now providing homo-oligomeric interactions that lock the E dimer with the fusion loop buried at its interface. Our data imply that during exocytosis of spiky immature particles, the hinged-lid pops open in the acidic environment of the TGN, creating a binding site for the pr moiety of an adjacent prM/E protomer. These additional interactions appear tailored to make tight $(prM/E)_2$ dimers, which in turn make lateral contacts to form a herringbone-like icosahedral organization while exposing the prM furin cleavage site to generate pr and M. The inter-dimer affinity for these lateral contacts is reflected, for instance, in the crystal packing of sE dimers from Zika virus, with dimers packing via interactions that are very

similar to those made by the central dimer within each raft of the herringbone pattern[33]. pr remains bound to the particle as long as the pH is acidic because of electrostatic complementarity, as shown in Fig. 2d. Dissociation of pr at neutral pH upon virus release from the infected cell is concomitant with a loss of electrostatic repulsion between the 150-loop and the charged N-ter helix (Fig. 5) such that the lid closes and binds the adjacent fusion loop while expelling pr. The positively charged, authentic N-terminal end of E is thus an important piece of the pH switch control, providing for the required interactions of the 150-loop in the closed-lid form, and its repulsion to pop the lid open upon exposure to low pH. Our finding that the carbohydrate attached to Asn154 is ordered in the closed hinge and disordered in the open lid form implies both entropic and enthalpic contributions of the glycan in each state, suggesting that it plays a role in the interaction. Indeed, knocking out the glycan in DENV2 (attached to E Asn153 in the DENV2 aa sequence) resulted in a virus with a higher pH threshold for fusion[34]. Yet, non-glycosylated variants of West Nile, Zika, and YF viruses are infectious and circulate in nature. These observations suggest that the lid is functional independent of its glycosylation status, which may however affect vector transmission, virulence and pathogenicity[35].

The flavivirus fusion machinery thus evolved three consecutive modes of protection against premature membrane insertion of the fusion loop. In the infected cell ER lumen, the newly budded spiky particles display the fusion loop capped by the pr moiety within the prM/E protomers. In the second stage, in the TGN, the pr moiety plays an additional role in the acid-pH driven transition by inducing formation of $(prM/E)_2$ dimers, thereby further locking the fusion loop and driving formation of smooth immature particles, on which the prM furin site becomes exposed. The third stage occurs after furin cleavage and upon deprotonation in the extracellular milieu, which weakens the interactions of pr with the E dimer. The 150-loop lid then snaps into place to knock out pr and stabilize the E dimers during the extracellular journey of the mature virion. This transient snap-lock is ready to open upon exposure to low pH in the endosome of a new cell, thereby triggering the membrane fusion reaction. The strength of the snap-lock may vary among flaviviruses, as suggested by differences in the phenomenon of breathing and the transient exposure of the FL in mature virions[36,37].

Besides flaviviruses, the other known class II enveloped viruses also rely on an accompanying protein (AP) to heterodimerize with the fusion protein (FP) and protect its fusion loop from premature exposure. The best example is provided by the alphaviruses and by the hantaviruses, for both of which the structure of the AP/FP heterodimer is available, as well as its organization on virions[38–40]. These structures have shown that not only the class II FPs but also the APs of these two otherwise unrelated viruses are structural homologs, and that the APs make the same type of lateral interactions on the particle to stabilize the surface glycoprotein lattice of the respective virions[41]. Further structural analyses combining experimental structures and recent powerful artificial intelligence-based methods for tridimensional structure prediction (Alphafold2[38]) have indicated that the structural homology between APs extends to the other families of bunyaviruses having class II viruses, such as the *Phenuiviridae*, *Nairoviridae*, *Perybunyaviridae* and *Tospoviridae*[39], suggesting that the FP/AP complex derives from a common, ancestral fusion machinery. In contrast, the flavivirus AP, prM, is unrelated to the ancestral AP and has been incorporated into the fusion machinery most likely at a later time point. This observation is in line with the unique organization of the flavivirus surface lattice, stabilized by lateral interactions of FP dimers only[42–44] in contrast to the AP/AP interactions observed in the other class II viruses[41]. We show that the gene coding for pr appears to have

been acquired by horizontal transfer from a cellular gene, hijacked for chaperoning the fold of flavivirus E in the ER of the infected cell[45] and incorporated into virions as part of a trans-viral-membrane-anchored full-length AP. This protein then further evolved to become a major actor in the transitions induced by low-pH exposure in the Golgi network to finally release infectious virions. The HSP40 co-chaperoning machinery is very active in the ER[46], cooperating in particular with the HSP70 chaperone BiP[47] to help fold the flavivirus glycoproteins that accumulate before budding. Assistance in folding by cellular chaperones is thus a natural process, and at some point, during flavivirus evolution one of these chaperonins became part of the virus. Beyond the insight into the unique flavivirus evolutionary pathway, our data provide a missing link explaining the acid pH-dependent transition from a spiky to a smooth immature particle in the TGN, as well as its priming mechanism to become fusogenic only after release from the producer cell.

## Methods

**Protein production and purification**. Recombinant proteins were derived from TBEV strain Neudoerfl (GenBank accession number U27495). All antigens were expressed in Drosophila Schneider 2 (S2) cells (Invitrogen) with a C-terminal enterokinase cleavage site and a double-strep-tag.

The pT389-TBEV prMe-linker-sE plasmid contains the gene for prM (aa 1-129) containing a deletion in the furin cleavage site (aa Arg89) and lacking the transmembrane anchor, followed by a linker with the TEV cleavage site (GGGGGENLYFQGGGG), and the gene for the ectodomain of E (aa 1-400)[48]. The pT389-TBEV pr plasmid contains the gene for the pr part of prM (aa 1-88). To generate the TBEV prMe-linker-sE$^{W101D}$ plasmid a corresponding point mutation was introduced into the pT389-TBEV prMe-linker-sE plasmid by site-directed mutagenesis (GeneArt site-directed mutagenesis system, Invitrogen) according to the manufacturer's instructions.

The recombinant proteins were expressed in Drosophila S2 cells as described previously[48]. Briefly, cells were transfected with the different expression plasmids and a plasmid containing a blasticidin resistance gene for selection following the manufacturer´s instructions (Invitrogen). Resistant cells were transferred into serum-free medium (Lonza) and expression was induced by the addition of 1 mM CuSO4. After seven to ten days, the supernatant was harvested, clarified and concentrated by ultrafiltration (Vivaflow 200, 30 MWCO, Sartorius). The strep-tagged proteins were purified by affinity chromatography with Strep-Tactin columns (IBA GmbH) according to the manufacturer's instructions.

The protein concentrations were determined with the Pierce BCA Protein Assay (Thermo Fisher Scientific) following the manufacturer's protocol. Purity of the proteins was verified by Sodium dodecyl sulfate- polyacrylamide gel electrophoresis (SDS-PAGE) according to Laemmli.

All proteins were stored at −20 °C and further purified by size exclusion chromatography (SEC) using a Superdex 200 10/300 GL column (GE Healthcare Life Sciences).

**Multi-angle static light scattering-Size exclusion chromatography**. MALS studies were performed using a SEC Superdex 200 column (GE Healthcare) previously equilibrated with the corresponding buffer, see below. SEC runs were performed at 25 °C with a flow rate of 0.4 ml/min, protein injection concentration was 100 μg. Online MALS detection was performed with a DAWN-HELEOS II detector (Wyatt Technology, Santa Barbara, CA, USA) using a laser emitting at 690 nm. Online differential refractive index measurement was performed with an Optilab T-rEX detector (Wyatt Technology). Data were analyzed, and weight-averaged molecular masses (Mw) and mass distributions (polydispersity) for each sample were calculated using the ASTRA software (Wyatt Technology). Equilibration buffer for prMe-linker-sE and prMe-linker-sE$^{W101D}$ was 50 mM Tris-HCl 8.0 and NaCl 500 mM. Equilibration buffers for addressing the effect of pH for sE, pr and the sE:pr complex were the three-component buffers, 100 mM Tris-HCl, 50 mM MES, 50 mM sodium acetate and 150 mM NaCl, at pH 5.5 or pH 8. The sE:pr complex, in 1:2 molar ratio, were prepared by incubation in the corresponding three-component buffers. Buffer exchange was performed by extensive dialysis of the sample, 12 h stirring at 4 °C and two 500 ml buffer replacement in 10 kDa molecular weight cut-off dialysis membranes (Spectrum). SEC fractions of sE:pr complexes at pH 5.5 or 8 were further analyzed by Coomassie blue SDS-PAGE.

**Surface plasmon resonance analysis**. The affinity of the sE protein for the pr peptide was measured by surface plasmon resonance (SPR) using a Biacore T200 system (GE Healthcare Life Sciences) equilibrated at 25 °C. The carboxylic groups of a Series S CM5 sensor chip were activated for 10 min using a mix of N-Hydroxysuccinimide (NHS, 50 mM) and 1-ethyl-3-[3-(dimethylamino)propyl]-carbodiimide (EDC, 200 mM). The CM5 sensor chip was immobilized with pr

protein at 16 μg/ml in acetate pH 4 or 4.5. The 20 min pr injection was followed by deactivation with 1 M ethanolamine for 7 min, reaching a density of amine coupled pr of 1150 and 600 resonance units (RU), for both acetate conditions, respectively. To note, 1 RU corresponds to a mass distribution of about 1 pg/mm² of sensor. Eight concentrations of sE protein (2-fold dilutions ranging from 11 μM to 7.8 nM) were injected at 30 μl/min for 700 s for pH 5 to 6.5, 300 s for pH 7 to 7.5 and 170 s for pH 8. At the end of each cycle, the surfaces were regenerated by sequential 15 s injections of three-component buffer at pH 9. Experiments were performed in duplicate, using different three-component buffers 100 mM Tris-HCl, 50 mM MES, 50 mM sodium acetate, at pH 5, 5.5, 6, 6.5, 7, 7.5 or 8, with 150 mM NaCl and 0.2 mg/ml BSA at 25 °C[49]. The association profiles were fitted using the Biacore T200 evaluation software (GE Healthcare) assuming a 1:1 interaction between E and pr.

**Crystallization and structure determinations**. Despite the presence of a TEV site in the 14-aa linker of the prMe-linker-sE and prMe-linker-sE$^{W101D}$ constructs, the proteins were not cleaved and were concentrated in 50 mM Tris-HCl pH 8.0 and NaCl 500 mM, prior to crystallization.

For the pr/sE complex, the two proteins pr and sE were incubated at pH 8 during 2 h at room temperature, in presence of an excess of pr, and then dialyzed overnight with three buffer changes of 10 mM MES pH 5 and 100 mM NaCl. After dialysis, the complex was purified by SEC at pH 5.5 (100 mM Tris-HCl, 50 mM MES, 50 mM sodium acetate, pH 5.5) and selected fractions were further concentrated.

Crystallization trials were performed at 18 °C in sitting drops of 400 nL formed by mixing equal volumes of the purified protein and reservoir solution in 96-well Greiner plates, using a Mosquito robot. The crystals were then optimized manually in 24-well plates using 2 μL hanging drops or with robotized setups on 400 nL sitting drops. The crystallization and cryo-cooling conditions used for the structure determinations, the crystal space groups, and diffraction characteristics are listed in Supplementary Table 2.

X-ray data for the three samples were collected at SOLEIL synchrotron PX1 beamline (St Aubin, France) and at the European Synchrotron Radiation Facility (Grenoble, France) on beamlines ID23-1 and ID29. The data sets were indexed, integrated, scaled and merged using the programs XDS[50] and AIMLESS[51] from the CCP4 suite of programs[52]. For X-ray data without anisotropy, the high-resolution limits were determined using CC$_{1/2}$-based cutoffs of 30%[53]. Anisotropy diffraction was measured for the prMe-linker-sE structure, the data were scaled and merged without applying a resolution limit. Then, DEBYE and STARANISO programs, developed by Global Phasing Ltd, were applied to the data using the STARANISO server (https://staraniso.globalphasing.com/cgi-bin/staraniso.cgi). These corrected anisotropic amplitudes were then used for further refinement of the structure prMe-linker-sE with BUSTER/TNT[54].

The three structures were determined by molecular replacement (MR) with the program PHASER[49] using as search models the pr protein from the YFV pr/sE complex (PDB 6EPK) and the TBEV sE protein (PDB 1SVB)[17] for the pr/sE and prMe-linker-sE structures, and the sE post-fusion (PDB 1URZ)[55] as a search model for the prMe-linker-sE$^{W101D}$ structure. The MR solutions were then followed by rigid-body refinement of each of the domains of sE and pr, then alternatively manually corrected using COOT[56] and refined using BUSTER/TNT[54]. TLS-based refinement[57] was performed for each structure. The final models were analyzed with MolProbity[58]. The refinement statistics are listed in Supplementary Table 2.

**Structural Analysis**. For visualization of the protein sequence variability, alignments representative of mammalian tick-borne flaviviruses were represented on the TBE pr/sE complex structure using the ConSurf server[48]. The polar contacts were computed with the 'Protein interfaces, surfaces and assemblies' service PISA at the European Bioinformatics Institute[59] (http://www.ebi.ac.uk/pdbe/prot_int/pistart.html). For the intermolecular interactions shown in Figures and Tables, the maximal cut-off distances used were 4 Å and 4.75 Å for polar and van der Waals contacts, respectively. The multiple sequence alignments were calculated using ClustalW[60] and displayed with ESPript (http://espript.ibcp.fr)[61]. The figures were prepared using the PyMOL Molecular Graphics System, version 2.0.0 (Schrödinger LLC) (http://pymol.sourceforge.net). Electrostatic surfaces were visualized in the program PyMOL using the APBS[62], PDB2PQR[63] and PROPKA[64] calculations software.

**Accession numbers**. Tick-borne encephalitis virus (strain Neudoerfl) (GenBank accession number U27495); Turkish sheep encephalitis virus (TSEV) (GenBank accession number DQ235151); Greek goat encephalitis virus (GGEV) (GenBank accession number DQ235153); Louping ill virus (LIV) (GenBank accession number NC_001809); Spanish sheep encephalitis virus (SSEV) (GenBank accession number DQ235152); Omsk hemorrhagic fever virus (OHFV) (GenBank accession number AAP29989); Langat virus (LGTV) (GenBank accession number AF253419); Alkhurma hemorrhagic fever virus (AHFV) (GenBank accession number AF331718); Kyasanur Forest disease virus (KFDV) (GenBank accession number JF416959); Powassan virus (strain LB) (POWV) (GenBank accession number L06436); Deer tick virus (DTV) (GenBank accession number AF311056); Royal Farm virus (RFV) (GenBank accession number DQ235149); Gadgets Gully virus (GGYV) (GenBank accession number DQ235145); Yellow fever virus (YFV) (GenBank accession number X03700); West Nile virus (WNV) (GenBank

accession number AF206518); Japanese encephalitis virus (JEV) (GenBank accession number AF315119); Kunjin virus (KUNJV) (GenBank accession number AY274504); Murray Valley encephalitis virus (MVEV) (GenBank accession number AF161266); Usutu virus (USUV) (GenBank accession number AY453412); St. Louis encephalitis virus (SLEV) (GenBank accession number DQ359217); Zika virus (GenBank accession number KJ776791); Dengue virus type 1 (DENV1) (GenBank accession number AB189120), type 2 (DENV2) (GenBank accession number M19197), type 3 (DENV3) (GenBank accession number AF349753), and type 4 (DENV4) (GenBank accession number AY618991).

**Reporting summary**. Further information on research design is available in the Nature Research Reporting Summary linked to this article.

## Data availability

The data that support this study are available from the corresponding authors upon reasonable request. Coordinates and structure factor amplitudes have been deposited in the Protein Data Bank under the accession numbers 7QRE (pr/sE), 7QRF (prMe-linker-sE) and 7QRG (prMe-linker-sE^W101D). The source data underlying Fig. 1a–e; Supplementary Fig. 1; and Supplementary Table 1 are provided as a Source Data file. Source data are provided with this paper.

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

## Acknowledgements

We thank the staff at beamlines PX1 and PX2 at SOLEIL synchrotron (St. Aubin, France) and at PX beamlines at the ESRF (Grenoble, France); A. Haouz and the staff at the crystallographic facility at Institut Pasteur; F. Agou from the Chemogenomic and Biological Screening platform at Institut Pasteur; P. Guardado Calvo and I. Fernandez for helping for the MALS experiments. This work was supported by the French ANR (Agence Nationale de la Recherche), grants ANR-13-ISV8-0002-01, ANR-10-LABX-62-10 IBEID, Wellcome Trust collaborative grant (UNS22082) to FAR, as well as by Institut Pasteur and CNRS. K.S and F.X.H acknowledge support from the Austrian Science Fund FWF (I1378-B13 and P27501-B21).

## Author contributions

F.X.H and F.A.R. conceived the project. I.M., G.T., K.S., and F.X.H. provided all the recombinant proteins. M.D. performed the SEC-MALS experiments and the crystal growing set-up on sE and pr. M.D. did the surface plasmon resonance experiments on sE and pr, with the help of P.E. S.D. made the bioinformatic analyses on pr and DnaJ. A.R. conducted the SEC-MALS experiments on the prMe-linker-sEconstructs, grew the crystals and collected synchrotron data with the help of M.C.V. M.C.V. grew the crystals of prMe-linker-sE and pr/sE complex and collected synchrotron data. M.C.V. processed all the data, built, refined, and analyzed all the atomic models. M.C.V., M.D., K.S., F.X.H., and F.A.R. planned the experiments. F.A.R. wrote the paper with the help of M.C.V., M.D., K.S., and F.X.H.

## Competing interests

The authors declare no competing interests.
