## [Peer Review File · Nature Communications]

Evolution and activation mechanism of the flavivirus class II membrane-fusion machineryReviewers' Comments:

Reviewer #1:

Remarks to the Author:

The manuscript by Vaney and colleagues sheds new and unique light on the role of the flavivirus prM protein and its protective role covering the envelope (E) protein fusion loop. The authors have examined the prM and E proteins of tick-borne encephalitis virus at acidic pH using X-ray crystallography. The structure reveals that there is a loop (150-loop) in the E protein that moves under differing pH conditions. Low pH favors the binding of the prM protein into a positively charged pocket at the E protein dimer interface. Following furin cleavage of prM and exposure to a neutral pH environment, pr is expelled and the 150-loop moves in to protect the fusion loop. The authors also show that pr moiety of prM is related to the HSP40 cellular chaperonin and has binding features that are similar. This is a significant and original piece of research that is very well described. The findings are relevant across the flavivirus genus.

Minor comments:

Figure 1: provide the molar ratios for pr and sE in panel A.

Figure 2a: In the top view, the Calpha of pr is difficult to see. Could the authors improve the contrast or change transparency of the E protein surface representation?

Page 9; top paragraph: Can the authors discuss what role (if any) the glycan present on the 150-loop plays in the open lid versus closed lid conformations? Have they removed that glycan to determine whether the loop continues to function as predicted?

Page 9; second paragraph: The authors make the argument that the 150-loop plays a similar role in not only tick-borne flaviviruses but also the mosquito-borne flaviviruses. However, they do not discuss the observations that this loop has been shown to be dynamic with some flaviviruses having much greater movement (access) than others (for example, dengue virus versus Zika virus). This should be addressed. Again, on page 12 of the Discussion, they mention that "the 150-loop lid then snaps firmly into place" suggesting there is no further movement/dynamics. The data suggests that this is not true although one can understand that there is displacement of pr by the movement of the 150-loop.

Page 11; line 16: Can the authors explain the linkage between (prM/E)₂ dimers that they are describing and the formation of the herringbone organization of E dimers?

Reviewer #2:

Remarks to the Author:

In this manuscript, the authors have identified, reconstituted, and determined the X-ray crystal structure of Tick-borne Encephalitis Virus envelope protein E in complex with pr protein. They found that the authentic N-terminal end of E is important in ordering the 150-loop that mediates a pH-dependent shedding of the pr protein upon viral egress. They show that slightly acidic pH is necessary for 2-to-2 complex formation and that both charge repulsion and loop rearrangement modulate sequential maturation steps. The authors also describe the structural homology of pr with its pre-viral distant relative chaperonin protein.

Major comments:

In all, this is a very nicely written and performed study with impressive mechanistic detail.

Minor comments:

Page 4, line 1: dissociation implies release, whereas in this instance subunits in the trimer are

handing-off to neighboring subunits in a conformational change. They don't really come apart.

Page 5, line 7: "did not permit to conclude" has a grammar error.

Page 5, lines 12-18: There is a distinct drop in both the k_{on} and k_{off} rates with increasing pH. The right-ward shift of the curves in Fig. 1E from pH 5.0 to 6.5 suggests modest conformational or charge changes to the affinity and yet they reach a common saturation level. Above the protonation pH of histidine, the k_{on} is drastically reduced and therefore saturates at a much lower level, indicating the binding surface, or indeed the E dimer, might not exist in those conditions. This is glossed over in the text.

Page 4-5 "The SEC profile of the sE monomer is aberrant, eluting at a large volume corresponding to small molecules..." This statement is somewhat awkward and may only be understood by individuals who perform SEC regularly. Consider rewording.

POINT-BY-POINT ANSWERS TO THE REVIEWERS' COMMENTS

Reviewer #1 (Remarks to the Author):

The manuscript by Vaney and colleagues sheds new and unique light on the role of the flavivirus prM protein and its protective role covering the envelope (E) protein fusion loop. The authors have examined the prM and E proteins of tick-borne encephalitis virus at acidic pH using X-ray crystallography. The structure reveals that there is a loop (150-loop) in the E protein that moves under differing pH conditions. Low pH favors the binding of the prM protein into a positively charged pocket at the E protein dimer interface. Following furin cleavage of prM and exposure to a neutral pH environment, pr is expelled and the 150-loop moves in to protect the fusion loop. The authors also show that pr moiety of prM is related to the HSP40 cellular chaperonin and has binding features that are similar. This is a significant and original piece of research that is very well described. The findings are relevant across the flavivirus genus.

We thank the reviewer for recognizing the originality of our research, and for her/his favorable assessment of our manuscript.

Minor comments:

1. Figure 1: provide the molar ratios for pr and sE in panel A.

The pr:sE stoichiometry was 2:1 monomer:monomer. This was added to the legend of Figure 1 (line 596)

2. Figure 2a: In the top view, the Calpha of pr is difficult to see. Could the authors improve the contrast or change transparency of the E protein surface representation?

In the revised version, we enhanced the color contrast in the cartoon representation of pr in Fig. 2a to make it stand out better.

3. Page 9; top paragraph: Can the authors discuss what role (if any) the glycan present on the 150-loop plays in the open lid versus closed lid conformations? Have they removed that glycan to determine whether the loop continues to function as predicted?

We have not removed the glycan to determine if the loop continues to function as predicted, and it would be beyond the scope of this manuscript to test this. Our collaborators in the Screaton lab in Oxford have shown that dengue viruses are viable when knocking out the corresponding glycan in the 150 loop (Personal communication). There are also data in the literature showing that knocking out the glycan in DENV2 results in a virus with a higher pH threshold for membrane fusion (0.65 pH units, <https://doi.org/10.1006/viro.1993.1252>), indicating that the glycan does stabilize the pre-fusion dimer. The fact that we see an ordered glycan when the lid is in the “down” position indicates that the inter-protomer interactions it displays are responsible for its ordering, as it is totally disordered when the lid is up. This is another indirect observation.

The lack of carbohydrate attached to the 150 loop in several circulating flavivirus strains might be used as an indirect evidence for the function of the lid as not strictly depending on the attached glycan. We have modified the text to incorporate the above comments (see lines 262-266):

“Our finding that the carbohydrate attached to Asn154 is ordered in the closed hinge and disordered in the open lid form implies both entropic and enthalpic contributions of the glycan in each state, suggesting that it plays a role in the interaction. Indeed, knocking out the glycan in DENV2 resulted in a virus with a higher pH threshold³⁴. Yet, non-glycosylated variants of West Nile, Zika, and YF viruses are infectious and circulate in nature. These observations suggest that the lid is functional independent of its glycosylation status, which may however affect vector transmission, virulence and pathogenicity³⁵.”

4. Page 9; second paragraph: The authors make the argument that the 150-loop plays a similar role in not only tick-borne flaviviruses but also the mosquito-borne flaviviruses. However, they do not discuss the observations that this loop has been shown to be dynamic with some flaviviruses having much greater movement (access) than others (for example, dengue virus versus Zika virus). This should be addressed. Again, on page 12 of the Discussion, they mention that “the 150-loop lid then snaps firmly into place” suggesting there is no further movement/dynamics. The data suggests that this is not true although one can understand that there is displacement of pr by the movement of the 150-loop.

The reviewer appears to refer to the dynamic “breathing” of flavivirus particles, in which the buried fusion loop is transiently exposed and can be accessible to antibodies targeting the fusion loop. Such breathing will also affect the interaction of the 150 loop with the fusion loop, since it involves E dimer dissociation. But to comply with the reviewer’s comments, we have eliminated the qualifier “firmly” in the sentence above, to say only “snaps into place” (line 277), as it is true that it might be more or less “firmly” depending on the flavivirus.

We now discuss the observation that the extent of breathing can differ among flaviviruses, possibly reflecting differences in the strength of the snap-lock. We have modified the text at lines 280-282 to reflect our response to the reviewer. We write there:

“The strength of the snap-lock may vary among flaviviruses, as suggested by differences in the phenomenon of breathing and the transient exposure of the FL in mature virions^{36, 37}.”

5. Page 11; line 16: Can the authors explain the linkage between (prM/E)₂ dimers that they are describing and the formation of the herringbone organization of E dimers?

We have shown for Zika virus and for dengue virus 2 (DENV2) that the E dimers have a natural affinity to pack laterally as in the rafts in the herringbone pattern. In the crystals of sE, sE dimers make extended rows of dimers packing in this way

(<https://doi.org/10.1016/j.cell.2021.11.010>). So once pr stabilizes E dimers at acidic pH, we postulate that this lateral affinity will lead to lateral packing of the dimers. The curvature of the virus particle does not allow to make rows of more than three E dimers, giving rise to the herringbone pattern. We have added a sentence in this paragraph, and a reference to the publication above (lines 253-256): “The inter-dimer affinity for these lateral contacts is reflected, for instance, in the crystal packing of sE dimers from Zika virus, with dimers packing via interactions that are very similar to those made by the central dimer within each raft of the herringbone pattern³³. ”

Reviewer #2 (Remarks to the Author):

In this manuscript, the authors have identified, reconstituted, and determined the X-ray crystal structure of Tick-borne Encephalitis Virus envelope protein E in complex with pr protein. They found that the authentic N-terminal end of E is important in ordering the 150-loop that mediates a pH-dependent shedding of the pr protein upon viral egress. They show that slightly acidic pH is necessary for 2-to-2 complex formation and that both charge repulsion and loop rearrangement modulate sequential maturation steps. The authors also describe the structural homology of pr with its pre-viral distant relative chaperonin protein.

Major comments:

In all, this is a very nicely written and performed study with impressive mechanistic detail.

We thank the reviewer for this very positive assessment of our work.

Minor comments:

1. *Page 4, line 1: dissociation implies release, whereas in this instance subunits in the trimer are handing-off to neighboring subunits in a conformational change. They don't really come apart.*

Yes, the protomers remain membrane-anchored to the viral particle, and so do not come apart in this sense. Yet, in order to make the dimers, the (prM/E)₃ trimers must dissociate from each other to allow re-formation of (prM/E)₂ dimers. Because of the confusion this sentence may cause, we have reworded it into (lines 68-72):
“Subsequent maturation into infectious particles during transport to the cell surface involves an acid-pH-induced reorganization of the 180 prM/E protomers at the particle surface. From forming 60 trimeric spikes, the protomers rearrange to form 90 head-to-tail dimers interacting laterally to make a smooth particle with an icosahedral herringbone-like arrangement^{25, 26}”

2. Page 5, line 7: “did not permit to conclude” has a grammar error.

Indeed. This was corrected. We actually changed the sentence to now read (lines 90-93): “The sE monomer elutes late from the SEC column, in fractions normally corresponding to the elution of small molecules and not of proteins of its molecular mass (~50 kDa). The reason is most likely retention in the column by interactions of the fusion loop with the resin, as observed previously with other class II fusion proteins²⁹”. This change also addresses the last comment raised by the reviewer (comment 4 below).

3. Page 5, lines 12-18: There is a distinct drop in both the k_{on} and k_{off} rates with increasing pH. The right-ward shift of the curves in Fig. 1E from pH 5.0 to 6.5 suggests modest conformational or charge changes to the affinity and yet they reach a common saturation level. Above the protonation pH of histidine, the k_{on} is drastically reduced and therefore saturates at a much lower level, indicating the binding surface, or indeed the E dimer, might not exist in those conditions. This is glossed over in the text.

We thank the reviewer for this comment. We have incorporated these remarks into the new version (lines 107-114):

“Figure 1e shows a right-ward shift of the curves from pH 5.0 to 6.5, suggesting modest conformational or charge changes to the affinity,

yet they reach a common saturation level. Above 6.5 (the protonation pH of histidine), the association rate is drastically reduced and therefore there is saturation at a much lower level, indicating that the binding surface might not exist under those conditions despite the presence of an sE dimer at neutral pH. In conclusion, the drastic drop in binding affinity upon raising the pH in the explored range suggests an electrostatic and/or a conformational change at the sE-pr interface upon deprotonation.”

4. Page 4-5 “The SEC profile of the sE monomer is aberrant, eluting at a large volume corresponding to small molecules...” This statement is somewhat awkward and may only be understood by individuals who perform SEC regularly. Consider rewording.

As requested, and as pointed out in the answer to point 2 raised by the reviewer, we have reworded the sentence into (lines 68-72):
“The sE monomer elutes late from the SEC column, in fractions normally corresponding to the elution of small molecules and not of proteins of its molecular mass (~50 kDa). The reason is most likely retention in the column by interactions of the fusion loop with the resin, as observed previously with other class II fusion proteins²⁹”